# Protective Effects of Alpha-Lipoic Acid against 5-Fluorouracil-Induced Gastrointestinal Mucositis in Rats

**DOI:** 10.3390/antiox11101930

**Published:** 2022-09-28

**Authors:** Deniz Ceylanlı, Ahmet Özer Şehirli, Sevgi Gençosman, Kerem Teralı, Hüseyin Şah, Nurhayat Gülmez, Serkan Sayıner

**Affiliations:** 1Department of Biochemistry, Faculty of Veterinary Medicine, Near East University, 99138 Nicosia, North Cyprus, Turkey; 2Department of Pharmacology, Faculty of Dentistry, Near East University, 99138 Nicosia, North Cyprus, Turkey; 3Department of Medical Biochemistry, Faculty of Medicine, Cyprus International University, 99258 Nicosia, Northern Cyprus, Turkey; 4Department of Histology and Embryology, Faculty of Veterinary Medicine, Near East University, 99138 Nicosia, North Cyprus, Turkey; 5Department of Histology and Embryology, Faculty of Veterinary Medicine, Siirt University, 56100 Siirt, Turkey

**Keywords:** 5-fluorouracil, alpha-lipoic acid, oxidative stress, gastrointestinal mucositis, inflammation

## Abstract

Alpha-lipoic acid (ALA) is extensively utilized in multivitamin formulas and anti-aging products. The purpose of this study was to investigate the potential protective benefits of ALA on 5-fluorouracil (5-FU)-induced gastrointestinal mucositis in Wistar albino rats. Tissues from the stomach, small intestine, and large intestine were excised, and blood sera were obtained to identify biochemical indices such as TNF-α, IL-1β, MDA, GPx, SOD, MMP-1, -2, -8, and TIMP-1. A histopathological study was also performed. The results revealed mucositis-elevated TNF-, IL-1, MDA, MMP-1, -2, -8, and TIMP-1 levels in both tissues and sera, and these values dropped dramatically following ALA treatment. Reduced SOD and GPx activities in mucositis groups were reversed in ALA-treated groups. The damage produced by mucositis in the stomach and small intestine regressed in the ALA-treated group, according to histopathological evaluation. Consequently, the implementation of ALA supplementation in 5-FU therapy may act as a protective intervention for cancer patients with gastrointestinal mucositis. In light of the findings, ALA, a food-derived antioxidant with pleiotropic properties, may be an effective treatment for 5-FU-induced gastrointestinal mucositus, and prevent oxidative stress, inflammation, and tissue damage in cancer patients receiving 5-FU therapy.

## 1. Introduction

Alpha-lipoic acid (ALA) is present in a wide range of foods, including vegetables (spinach, broccoli, tomatoes) and meats (especially viscera), as well as several nutritional supplements [1,2]. ALA has numerous antioxidant effects, including the activation of endogenous antioxidant mechanisms and the chelation of metal ions such as iron, zinc, and copper [3,4]. As an organosulfur molecule, ALA can scavenge reactive oxygen species (ROS) and boost the activity of tissue antioxidant enzymes such as glutathione peroxidase (GPx) and superoxide dismutase (SOD) [5,6,7]. Because of its antioxidant properties, ALA has been found to reduce ALA-dependent cell death in lung cancer [8], breast cancer [9], and colon cancer [10]. Furthermore, earlier research has shown that ALA possesses anti-inflammatory properties [11,12,13,14,15,16]. ALA works to reduce inflammation by modulating the expression of inflammatory cytokines such as tumor necrosis factor-alpha (TNF-α) and interleukin-1β (IL-1β) [17,18]. The pharmacological benefits of ALA have been proven in a wide variety of medicinal-related fields, including cardiovascular, neuroprotective, cognitive, and anti-aging models [19].

Cancer is the leading cause of mortality in every country and a major impediment to longevity [20]. The side effects of chemotherapeutic medications are one of the most important debilitating factors in the fight against cancer, and mucositis is one of the most prevalent diseases encountered in cancer patients [21]. Mucositis is a pathological inflammatory condition characterized by ulceration and inflammation of the mucosa of the gastrointestinal system [22]. Mucositis increases the use of opioid analgesics due to pain and quickens the transition to total parenteral nutrition due to nutritional issues [23]. Mucositis and mucositis-related difficulties diminish the patients’ quality of life, while the development of life-threatening conditions and negative responses to therapy are a cause for concern due to poor lesion treatment [24].

The most often used chemotherapeutic drug in gastrointestinal malignancies is the pyrimidine analogue 5-fluorouracil (5-FU). 5-FU exerts its anticancer effects by inhibiting thymidylate synthase (TS) and integrates its metabolites into RNA and DNA [25]. However, this agent induces gastrointestinal mucositis, a serious debilitating side effect [26]. In particular, the degradation of the oxidant-antioxidant balance, a rise in pro-inflammatory cytokine levels, and proteolytic enzyme activity, play a major role in the development of these adverse effects. Although antioxidant and anti-inflammatory medicines have been utilized in an attempt to address this issue, a particularly robust therapeutic method has yet to be identified [27,28].

To date, several studies have reported the use of 5-FU and ALA together in different cancers; these include cervical and colorectal tumors [29,30]. Furthermore, auxiliary medications that might be incorporated into treatment procedures to prevent cancer patients from the gastrointestinal mucositis adverse effects induced by 5-FU would be advantageous. Thus, the purpose of this study was to determine the biological effects of using ALA in rats with a mucositis model created by 5-FU, in addition to examining its possible use as a candidate agent in treatment protocols.

## 2. Materials and Methods

### 2.1. Animals

The animals were taken from the Animal Experiments Units of the Near East University. The rats (Wistar albino, both sexes) were maintained under a 12:12 h dark/light cycle from the start at a room temperature of 22  ±  2 °C in a humidity-controlled facility (50 ± 5%). Feeding was ad libitum without any restriction to their standard rat chow and drinking water. Rats were housed in a plexiglass cage (4 rats per cage with dimensions of 60  ×  40  ×  40 cm). The study protocol was authorized by the Near East University Local Animal Experiments Ethics Committee (Approval No. 2019/01 on 17 January 2019 and 2020/11 on 27 November 2020), and ethical guidelines mentioned in the 1964 Declaration of Helsinki and its later amendments were followed.

Thirty-two Wistar Albino rats of both sexes weighing 200–250 g were allocated into four groups of eight rats each (*n* = 8): Control, ALA, Mucositis, and Mucositis + ALA groups. The Control group received only intraperitoneal (i.p.) saline (SF) once a day for one week. The ALA group received only ALA at a dose of 100 mg/kg i.p. once a day for one week from day 1 [31,32]. The Mucositis group was treated with SF (i.p.) on day 1 and 400 mg/kg 5-FU i.p. from day 2 to the end of the experiment [33]. The Mucositis + ALA group received 100 mg/kg i.p. ALA on day 1 and 400 mg/kg i.p. 5-FU on day 2. Then, 100 mg/kg/day i.p. ALA administration was continued until the end of the experiment. The animals were monitored for one week with no mortalities experienced during that time. The animals in all groups were euthanized at the end of the research period by the administration of a combination of a ketamine/xylazine overdose. Laboratory assays were performed at the Diagnostic Laboratory and Histopathology Laboratory of the Animal Hospital, Near East University.

### 2.2. Collection of Blood and Tissue Samples

Blood samples were obtained and submitted to the laboratory in serum separator tubes. The serum samples were separated at 2000× *g* for 10 min and kept at −80 °C until further analysis. To eliminate extra blood, stomach, small intestine, and large intestine tissues were collected and rinsed with phosphate buffer solution (pH = 7.4). Then, after being delivered to the laboratory, they underwent a homogenization procedure. Tissue homogenization was performed according to the manufacturer’s protocol using RIPA buffer (item no. 10010263, Batch No. 0490889-1, Cayman Chemicals, Ann Arbor, MI, USA) and Dounce tissue grinder set (D8938, Lot. 3110, Sigma-Aldrich, St. Louis, MO, USA) on ice. After homogenization, samples were centrifuged at 1600× *g* for 10 min at +4 °C. The Bradford protein determination technique was utilized to quantify protein levels in tissue samples, and supernatants were then frozen at −80 °C for subsequent investigation.

### 2.3. Measurements of Biochemical Indices

Activities of alanine transaminase (ALT), aspartate transaminase (AST), alkaline phosphatase (ALP), lactate dehydrogenase (LDH), lipase (LIP), amylase (AMY), and creatinine, blood urea nitrogen (BUN), total protein (TP), and albumin levels were measured as biochemical indices in serum samples using an automated BS-240 VET Clinical Chemistry Analyser (Mindray, Shenzhen, China).

### 2.4. Measurement of Cytokines in Sera and Tissues

TNF-α and IL-1β concentrations in sera and tissue samples were measured using commercially available rat-specific enzyme-linked immunosorbent assay (ELISA) assay kits (ELR-TNF-α and ELR-IL1β-, RayBiotech Life Inc., Norcross, GA, USA) following manufacturer’s instructions.

### 2.5. Measurement of Matrix Metalloproteinases (MMPs) and Tissue Inhibitors of Metalloproteinase-1 (TIMP-1) in Sera and Tissues

MMPs and TIMP-1, which are involved in a variety of biological processes, such as cancer and inflammation, were measured in both tissue homogenates and sera using the manufacturer’s instructions and recommendations of commercially available rat-specific ELISA assays (MMP-1 E-EL-R0617; Elabscience, Wuhan, China; MMP-2 ELR-MMP2; MMP-8 ELR-MMP8; TIMP-1 ELR-TIMP-1 RayBiotech Life Inc., Norcross, GA, USA). Serum concentrations of the MMPs and TIMP-1 were expressed as pg/mL, while tissue levels were expressed as pg/mg protein.

### 2.6. Measurement of Malondialdehyde (MDA), GPx and SOD in Tissues

MDA levels in tissues were evaluated using commercially available assay kits to determine the lipid peroxidation state (TBARS Assay Kit, item no. 10009055, Cayman Chemicals, Ann Arbor, MI, USA). The measuring principle is based on the reaction with thiobarbituric acid (TBA) in boiling water for 60 min in an acidic medium and the measurement of the absorbance of the reaction mixture at 532 nm (Ohkawa et al., 1979). A VersaMax Tunable Microplate Reader was used to measure the absorbance (Molecular Devices, San Jose, CA, USA). MDA concentrations in tissues were expressed as nmol MDA/mg protein.

In tissues, GPx and SOD activity were evaluated using ready-to-use assay kits. (RANSEL RS505, RANSOD SD125, Randox Laboratories Ltd., Crumlin, UK) by using an automated BS-240 VET Clinical Chemistry Analyser (Mindray, Shenzhen, China). The quantities of GPx and SOD in the tissue samples (stomach, small intestine, and large intestine) were expressed as U/mg protein.

### 2.7. Histopathological Evaluation

The tissues of the stomach, small intestine, and large intestine obtained immediately after the experimental period were fixed in 10% (*v*/*v*) neutral formalin for 24 h at room temperature. After fixation, the tissues were placed on a tissue tracking device, Leica TP1020 (Leica Microsystems GmbH, Wetzlar, Germany). Tissues taken from the tissue tracking device were then embedded in paraffin. Sections of 5-μm thickness were cut from the tissues with the help of a microtome Leica RM2255 (Leica Microsystems GmbH, Wetzlar, Germany), and routine hematoxylin & eosin (HE) staining was performed. Sections were examined histomorphologically with a Leica DM500 light microscope coupled with the Leica Microsystem Framework integrated digital imaging analysis system (Leica Application Suite version 3.0 Series 38132019, Leica ICC50 HD, Leica Biosystems Nussloch GmbH, Nußloch, Germany).

### 2.8. Statistical Analyses

GraphPad Prism software was used for statistical analysis (version 7.04, GraphPad Software, San Diego, CA, USA). The data were all expressed as mean ± standard deviation ($\bar{X}$ ± SD). An analysis of variance (ANOVA) test was used to compare data groups, followed by Tukey’s multiple comparison test. In addition, the Kruskal–Wallis test was used to compare histomorphological scores among groups. A difference of *p* < 0.05 was considered statistically significant.

## 3. Results

### 3.1. ALA Positively Affects Biochemical Indicies

Table 1 displays serum enzyme activity and metabolite concentrations. When compared to the Control group, the Mucositis group had substantially higher levels of albumin, ALT, AST, ALP, LDH, amylase, lipase, BUN, creatinine, and TP (*p* < 0.05–0.0001). However, there was a significant difference in albumin, AST, ALT, ALP, LDH, BUN, creatinine, amylase, BUN, and creatinine levels between the ALA + Mucositis group and the Mucositis group (*p* < 0.05–0.001). ALP, ALT, AST, LDH, and amylase levels were also substantially different between the Mucositis and ALA groups (*p* < 0.05–0.0001). Accordingly, the levels of liver enzymes, pancreatic enzymes, and metabolites were observed to be dramatically reduced in the Mucositis group when ALA was added to the treatment.

### 3.2. ALA Decreases the Production of Pro-Inflammatory Cytokines

The levels of pro-inflammatory cytokines TNF-α and IL-1β in serum, stomach, small intestine, and large intestine samples can be seen in Figure 1. The TNF-α and IL-1β levels in sera and all tissues of animals in the Mucositis group were significantly higher when compared to Control, ALA and Mucositis + ALA groups (*p* < 0.05–0.0001). However, TNF-α and IL-1β levels in the sera and all tissues reduced significantly in the Mucositis + ALA group compared to the Mucositis group (*p* < 0.05–0.0001). Between the Control and Mucositis + ALA groups, there was a substantial difference in the levels of TNF-α in the small and large intestines. (*p* < 0.01). There were also significant differences in TNF-α levels in the large and small intestines between the ALA and Mucositis + ALA groups. (*p* < 0.01, *p* < 0.001, respectively). Furthermore, there were statistically significant differences between the Control and Mucositis + ALA groups in terms of IL-1 levels in the stomach (*p* < 0.05). Therefore, ALA appeared to significantly suppress the levels of pro-inflammatory cytokines in mucositis.

### 3.3. ALA Treatment Reduces Lipid Peroxidation and Increases the Activities of Antioxidant Enzymes in Tissues

MDA levels, GPx and SOD activities were quantified to assess peroxidation and antioxidant status at the tissue level (Figure 2). MDA levels in the Mucositis group’s stomach, small intestine, and large intestine were significantly higher than in the Control, ALA, and Mucositis + ALA groups (*p* < 0.05–0.0001). In addition, the levels of MDA in the large intestines of the Mucositis + ALA group were substantially different from those of the Control and ALA groups (*p* < 0.0001).

Rats in the Control and ALA groups had significantly higher SOD levels in all tissues than rats in the Mucositis group (*p* < 0.05–0.0001). SOD levels in the large intestine samples were significantly different in the ALA and Mucositis + ALA groups (*p* < 0.01). In the stomach samples, the SOD levels were significantly lower in the ALA and Mucositis + ALA groups (*p* < 0.05). Furthermore, SOD levels in the small intestine of rats from the Mucositis group were substantially lower than SOD levels in the ALA group (*p* < 0.05). However, SOD levels in the stomach and large intestine of animals in the Mucositis and Mucositis + ALA groups were substantially different (*p* < 0.05–0.0001).

### 3.4. ALA Treatment Increases the Levels of Tissue MMPs and TIMP-1

MMP-1, MMP-2, MMP-8, and TIMP-1 activities (Table 2) in serum, stomach, small intestine, and large intestine were significantly higher in the Mucositis group in comparison to the Control and ALA groups (*p* < 0.05–0.0001). The increase in MMP-1, MMP-2, MMP-8, and TIMP-1 levels in Mucositis rats’ sera and stomach, small intestine and large intestine were significantly suppressed in Mucositis + ALA rats (*p* < 0.05–0.0001), although there was no significant difference in serum MMP-8 levels between the Mucositis + ALA and Mucositis groups.

### 3.5. Histological Evaluation of Stomach, Small and Large Intestines

Surface epithelial cells, foveola gastrica sections, gland epithelial cells, submucosa, muscularis, and serosa layers were all normal in the Control and ALA groups’ gastric sections (Figure 3a,b). Degeneration of the surface epithelial cells, edema, and extensive gland enlargement were seen in the Mucositis group (Figure 3c). Degeneration in epithelial cells was observed as local foci. Mild congestion between the glands was noted in one of the subjects in this group. Degeneration of epithelial cells, edema, and gland enlargement were observed to be less evident in the Mucositis + ALA group (Figure 3d).

An ocular micrometre was used to quantify villus heights in the small intestine. Although the villi lengths in the mucositis group were decreased compared to the other groups, no statistical difference was found between the groups (*p* > 0.05) (Table 3). The tunica mucosa, submucosa, muscularis, and serosa layers exhibited normal histological structure in the small intestine of the Control and ALA groups (Figure 4a,b). Irregularities in the villi and shedding of enterocytes were observed in the Mucositis group (Figure 4c). Villus irregularities were found to be decreased in the Mucositis + ALA group (Figure 4d). 

**Figure 3 antioxidants-11-01930-f003:**
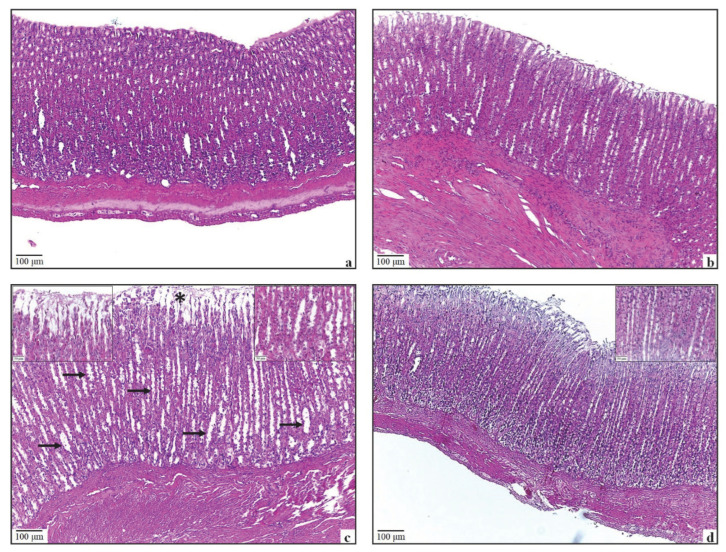
Histological view of HE stained stomach tissue for (**a**) the Control group, (**b**) the ALA group, (**c**) the Mucositis group, and (**d**) the Mucositis + ALA group. Bar = 100 µm; enlarged area Bar = 50 µm. (**a**) It is seen that epithelial cells and gland structures of the corpus-fundus region of the stomach of the control group are normal. (**b**) The stomach structure of the ALA-only group is similar to the control group. (**c**) Degeneration of gastric epithelial cells and edema (*) and enlargement of glands (arrows) are common in the Mucositis group. (**d**) It is seen that edema in the surface epithelial cells and enlargement of the glands decreased in the Mucositis + ALA group compared to the Mucositis group. In both the control and experimental groups, no structural alterations in the large intestine were seen. Visually normal simple layered columnar epithelial cells, goblet cells, and glands were observed (Figure 5a–d).

## 4. Discussion

Cancer incidence and death rates are rising dramatically across the world, reflecting both a growing and an aging population, together with changes in the prevalence and distribution of the main cancer risk factors, many of which are linked to socioeconomic development [34,35]. Chemotherapy goes back several centuries, however, the first significantly successful and scientifically documented use of chemotherapy for cancer systemically did not occur until the 1940s. The first successful treatment method applied to cancer is based on the experience of combating the toxic effects of nitrogen mustard on body systems, and later, it was used in the treatment of cancer patients. Although there were rapid tumor relapses following a significant antitumor effect, it was initially seen as a positive outcome in treatment. This experience appeared to be a clue to the initiation of chemotherapy for malignant tumors [36].

5-FU is a commonly used in anticancer medication since its inception in 1957 and has played a significant role in the treatment of colon cancer. It is also utilized in patients with other malignancies, such as breast and head and neck cancers [37]. The nucleoside metabolism and its incorporation into RNA and DNA results in variable toxicity and subsequent tissue damage [38]. Various mechanisms have been suggested for the toxic effects of the anti-metabolite drug 5-FU, such as causing oxidant-antioxidant imbalance, increasing cytokine levels and proteolytic enzyme activity and eventually activating different mechanisms of tissue damage [39,40].

Anticancer medications such as 5-FU may induce a significant decrease in antioxidant enzyme activity, such as SOD and GPx, as well as harmful impacts on non-enzymatic antioxidants, such as sulfhydryl groups. These findings fully support prior research indicating that 5-FU therapy is associated with oxidative imbalance [41,42,43]. Evidence from previous studies has reported that oxidative stress is responsible for the pathogenesis of 5-FU-induced gastrointestinal damage via excessive free radical release and ROS [44]. According to results from earlier studies, the pathophysiology of oxidative stress is caused by the excessive free radical release and ROS gastrointestinal damage caused by 5-FU [45,46,47]. Our findings reveal that 5-FU triggers lipid peroxidation and consequent damage to cellular membranes, in addition to causing a depletion of cellular SOD and GPx antioxidants. These findings agree with previous reports [47,48,49,50]. The most well-known activities of ALA promote its use as a possible therapeutic option for diseases related to oxidative stress. Furthermore, ALA is known to have anti-inflammatory characteristics which help to protect the gastrointestinal system. In our study, MDA levels increased in the stomach and small and large intestines after 5-FU administration, whereas SOD and GPx activities reduced, and these alterations were alleviated by ALA therapy. ALA has been shown to have exceptional antioxidant properties in several models, not only in stomach ulcers but also in the small and large intestines, through the removal of heavy metals that cause increased oxidative stress and by rebuilding the antioxidant defense system [51,52,53,54].

A high rate of destruction of the general architecture of the stomach and intestines is a known important feature of gastrointestinal mucositis, expressed mainly as rupture and atrophy of intestinal villi, loss of crypt structure, degeneration of gastric epithelial cells, increased goblet cell emptying, and infiltration of inflammatory cells [55]. The most noticeable feature of these is usually a deterioration in mucosal integrity and also alterations in villi and crypt parameters [56,57]. The effects of ALA on 5-FU-induced gastrointestinal mucositis were reported in this study, demonstrating that ALA could reverse the deleterious effects of an antineoplastic agent, 5-FU, on the intestines, including oxidative damage, neutrophil recruitment, mastocytosis, goblet cell depletion, and histological and morphometric alterations. Our results are also compatible with those found in previous studies [17,58,59,60]. ALA, which we use in our treatment protocol, has previously been shown to have protective effects on gastrointestinal tissues in both ulcer and colitis models [59,61,62]. In this study, gastrointestinal mucosal damage was structurally improved in groups treated with ALA.

This study showed that proinflammatory cytokines IL-1β and TNF-α are elevated in Mucositis group’s sera and tissues (the stomach, small intestine, or large intestine). Earlier studies have demonstrated that 5-FU administration increases plasma IL-1β and TNF-α levels, which are in line with our plasma findings [63,64]. However, we were unable to find any previously published reports examining the effects of these cytokines in the stomach and intestinal tissues. Therefore, our findings suggest that the effects of these cytokines contribute to structural damage not only via their systemic effects but also through their local impacts. It was observed that the increase in IL-1β and TNF-α levels in Mucositis rats can be reversed by ALA treatment. Intriguingly, ALA has been shown to suppress only plasma cytokine production in studies conducted to date [65,66]. Thus, it is thought that suppressing not only plasma cytokine levels but also tissue cytokine levels is likely to a represent pivotal mechanism underlying the protective activity of ALA against mucositis. 

The five subgroups of the MMP family of calcium-dependent zinc-containing enzymes include collagenases, stromelysins, gelatinases, membrane-type MMPs, and other endopeptidases [67]. They are expressed by epithelial, mesenchymal, and hematopoietic cells [68]. MMP-1, MMP-2, MMP-8, and TIMP-1 in particular have been proven in experimental and clinical investigations to influence inflammation [69,70,71]. Our study has demonstrated that gastrointestinal mucositis induced by 5-FU increases the activity of the proteolytic enzymes MMP-1, MMP-2, MMP-8, and TIMP-1 in sera and tissues. Therefore, by degrading extracellular matrix proteins, MMPs and TIMP-1 clearly exert damaging effects on tissues and are crucial to the pathophysiology of gastrointestinal mucositis.

In comparison to the Control and ALA groups, 5-FU treatment enhanced MMP-1, MMP-2, MMP-8, and TIMP-1 activation in sera and tissues in the Mucositis + ALA group. Several models of inflammation have been utilized to examine the effect of ALA on the expression levels of MMP-1, MMP-2, MMP-8, and TIMP [72,73]. ALA’s anti-inflammatory and antioxidant effects in inflammation models have revealed that the protective effect is produced by lowering MMP-1, MMP-2, and MMP-8 expression while enhancing TIMP-1 activity [74,75]. Several of the 26 recognized MMPs have been extensively studied in inflammatory diseases and cancer. MMP-8 and -9 are collagenase MMPs that are expressed by fibroblasts and infiltrating inflammatory cells in addition to tumor cells [76].

Although some studies indicate that TIMP-1 activation plays a protective role by inhibiting MMP activation, some other studies emphasize that increased TIMP-1 activation leads to an inflammatory response by increasing proinflammatory cytokine expression [77,78]. Enhanced TIMP-1 activation and increased production of proinflammatory cytokines in our study imply that this phenomenon is confirmed [79]. 

## 5. Conclusions

Chemotherapeutics, which are often advised as first-line treatment for cancer, pose a significant challenge due to their adverse side effects. 5-FU, which is used to treat malignancies of the breast, colon, rectum, stomach, and pancreas, has the notable side effect of causing gastrointestinal mucositis. Therefore, the use of medicines that may be provided alongside chemotherapeutics to avoid or decrease these adverse effects is critical, and this area of study is still under investigation. We feel that by using ALA in our research, we have contributed to the corpus of knowledge in this area. As a result of our research, we have discovered that the combination of ALA and 5-FU induces favorable changes in the parameters involved in the control of inflammation, together with an oxidant–antioxidant balance. Our findings indicate that the incorporation of ALA into 5-FU therapy is a potential option for cancer patients with gastroenteritis.

## Figures and Tables

**Figure 1 antioxidants-11-01930-f001:**
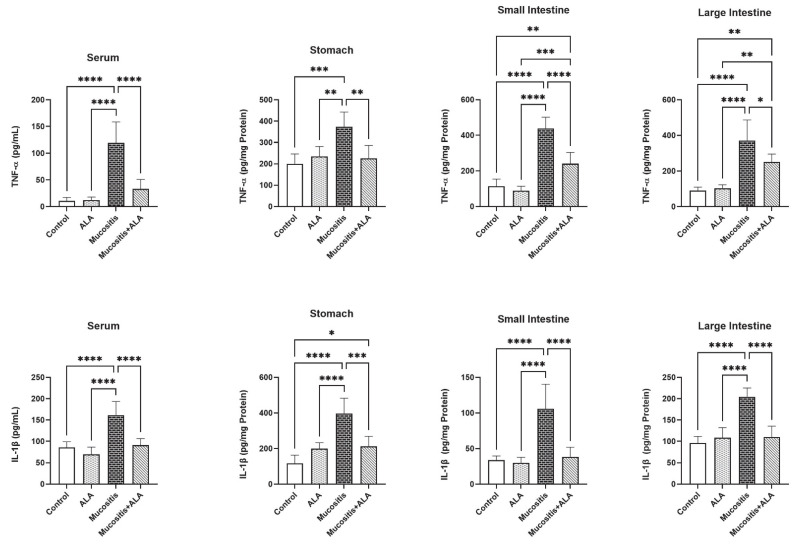
Comparisons of serum, stomach, small intestine, and large intestine concentrations of) TNF-α, IL-1β in the Control, ALA, Mucositis and Mucositis + ALA groups. * *p* < 0.05, ** *p* < 0.01, and *** *p* < 0.001 and **** *p* < 0.0001.

**Figure 2 antioxidants-11-01930-f002:**
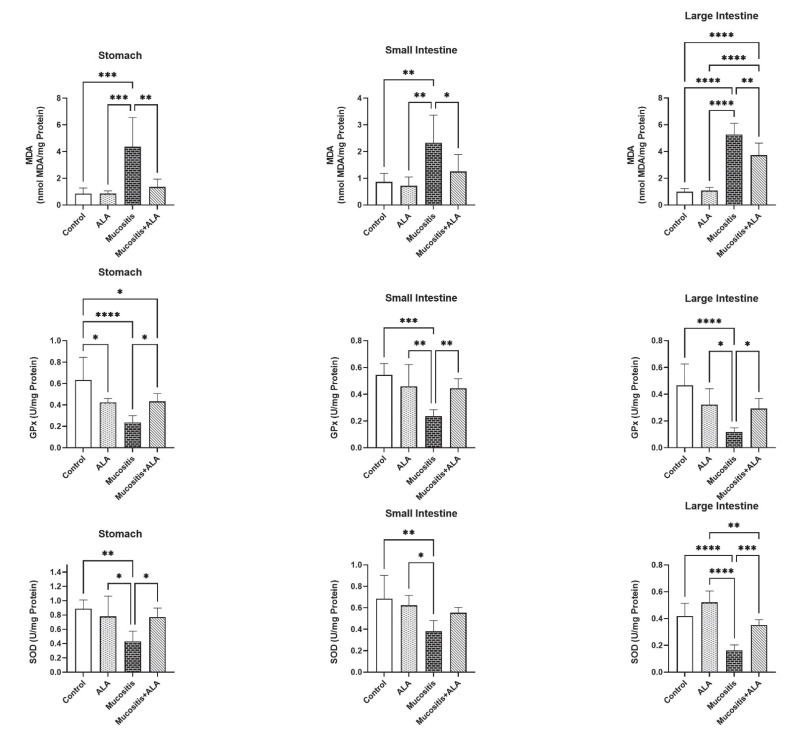
Comparisons of MDA levels and GPx and SOD activities in the stomach, small intestine, and large intestine in the Control, ALA, Mucositis and Mucositis + ALA groups. * *p* < 0.05, ** *p* < 0.01, and *** *p* < 0.001 and **** *p* < 0.0001. In Mucositis group rats, GPx levels in all tissues were significantly reduced compared to the Control, ALA and Mucositis + ALA groups (*p* < 0.05–0.0001). Also, in the stomach tissue, the GPx levels of animals in the ALA and Mucositis + ALA groups were shown to be significantly different to the Control group (*p* < 0.05).

**Figure 4 antioxidants-11-01930-f004:**
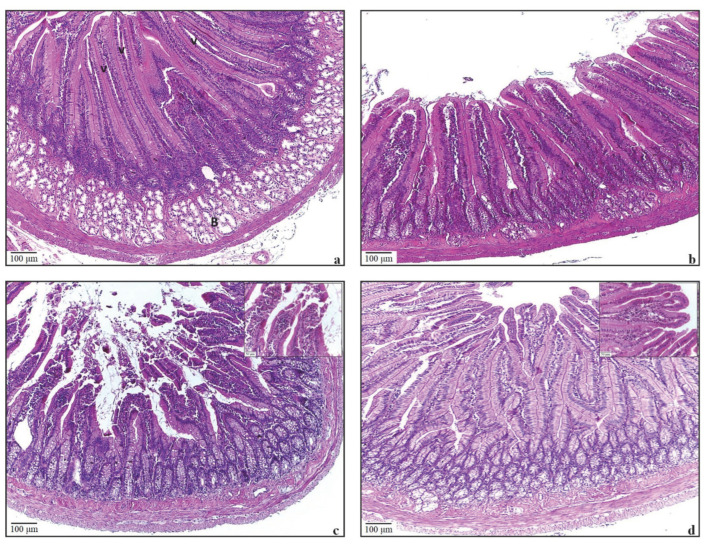
The microscopic images of HE-stained slides of the small intestine tissue. Bar = 100 µm; enlarged area Bar = 50 µm. (**a**) The structure of the small intestine of the control group is shown; Villus (v) and Brunner’s glands (B). (**b**) The structure of the small intestine in the ALA-only group is similar to the control group. (**c**) Shedding of villi and irregularity in enterocytes are seen in the Mucositis group. (**d**) In the Mucositis + ALA group, it is seen that the villi structures are smooth.

**Figure 5 antioxidants-11-01930-f005:**
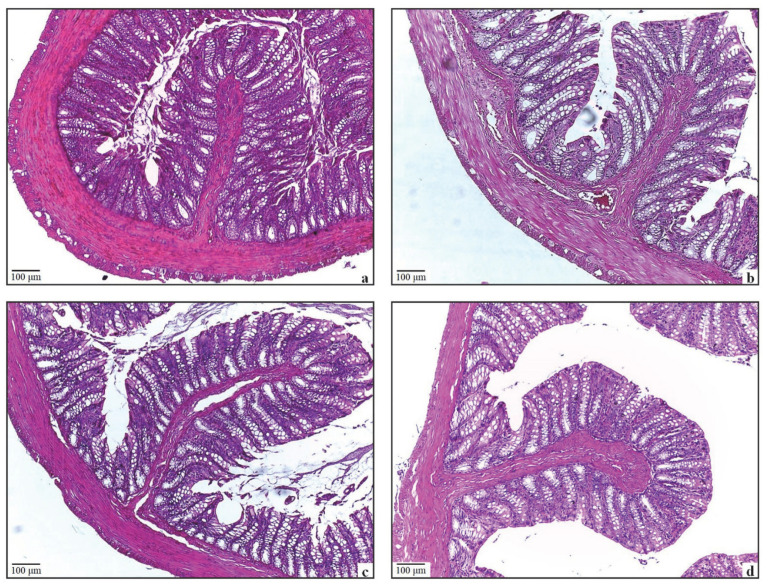
The microscopic images of HE-stained slides of the large intestine tissue. Bar = 100 µm. (**a**) The image shows a large intestine of the control group. (**b**) The structure in the ALA-only group is similar to the control group. (**c**) Goblet cell density in the Mucositis group was similar to the control and ALA groups, no degeneration was observed in enterocytes. (**d**) The structure in the Mucositis + ALA group is similar to the control group.

**Table 1 antioxidants-11-01930-t001:** Results of serum enzymes and metabolites in the sera of rats. Results are expressed as mean ± SD.

	Control	ALA	Mucositis	Mucositis + ALA
Albumin (g/dL)	2.97 ± 0.16	3.22 ± 0.35 *	3.76 ± 0.22 **	3.10 ± 0.57 ^§^
TP (g/dL)	4.44 ± 0.47	5.28 ± 0.93	5.77 ± 0.42 *	4.95 ± 0.87
ALP (U/L)	64.01 ± 21.15	71.85 ± 19.19	120.50 ± 11.97 ***^,††^	69.14 ± 24.45 ^§§^
ALT (U/L)	27.36 ± 9.12	25.00 ± 4.73	43.55 ± 11.10 **^,††^	25.57 ± 3.85 ^§§^
AST (U/L)	109.6 ± 13.2	114.3 ± 5.3	142.3 ± 13.2 **^,†^	118.4 ± 19.4 ^§^
LDH (U/L)	939 ± 249	681 ± 156	1853 ± 454 ***^,††††^	1241 ± 377 ^§^
BUN (mg/dL)	18.04 ± 2.29	21.49 ± 2.71	23.80 ± 1.49 ***	18.22 ± 1.90 ^§§^
Creatinine (mg/dL)	0.30 ± 0.09	0.41 ± 0.86	0.51 ± 0.048 ***	0.37 ± 0.040 ^§^
Amylase (U/L)	1350 ± 165	1681 ± 188	2575 ± 426 ****^,†††^	1659 ± 312 ^§§§^
Lipase (U/L)	24.50 ± 4.84	27.67 ± 4.22	31.83 ± 3.97 *	27.67 ± 4.54

* *p* < 0.05 ** *p* < 0.01 *** *p* < 0.001 **** *p* < 0.0001 compared to Control; ^†^
*p* < 0.05 ^††^
*p* < 0.01 ^†††^
*p* < 0.001 ^††††^
*p* < 0.0001 compared to ALA; ^§^
*p* < 0.05 ^§§^
*p* < 0.01 ^§§§^
*p* < 0.001 compared to Mucositis.

**Table 2 antioxidants-11-01930-t002:** Results of MMPs and TIMP-1 in sera and tissues (Mean ± SD).

	Control	ALA	Mucositis	Mucositis + ALA
MMP-1 (sera pg/mL; tissues pg/mg protein)				
Serum	1.09 ± 0.18	1.14 ± 0.37	2.99 ± 0.81 ***^,†††^	1.87 ± 0.87 ^§^
Stomach	1.01 ± 0.29	1.03 ± 0.19	1.94 ± 0.36 ****^,†††^	1.34 ± 0.27 ^§§^
Small Intestine	0.68 ± 0.23	0.99 ± 0.41	2.36 ± 0.67 ****^,†††^	1.35 ± 0.23 ^§§^
Large Intestine	0.78 ± 0.14	0.60 ± 0.15	1.26 ± 0.22 ***^,††††^	0.96 ± 0.18 ^§^
MMP-2 (sera pg/mL; tissues pg/mg protein)				
Serum	25.06 ± 6.36	23.74 ± 3.37	37.13 ± 7.81 **^,††^	20.04 ± 4.14 ^§§§^
Stomach	1.39 ± 0.32	1.45 ± 0.27	2.07 ± 0.27 **^,††^	1.59 ± 0.31^§^
Small Intestine	0.74 ± 0.33	0.86 ± 0.19	1.38 ± 0.12 ***^,††^	0.98 ± 0.24 ^§^
Large Intestine	0.73 ± 0.20	0.81 ± 0.21	1.39 ± 0.29 ***^,††^	0.86 ± 0.20 ^§§^
MMP-8 (sera pg/mL; tissues pg/mg protein)				
Serum	61.67 ± 20.49	73.23 ± 14.00	176.40 ± 110.00 *^,†^	111.00 ± 34.25
Stomach	2.66 ± 0.45	2.18 ± 0.59	4.89 ± 2.26 *^,††^	2.26 ± 0.74 ^§§^
Small Intestine	0.95 ± 0.31	0.69 ± 0.36	1.98 ± 0.55 **^,††††^	1.06 ± 0.32 ^§§^
Large Intestine	0.60 ± 0.22	0.65 ± 0.29	1.74 ± 0.47 ****^,††††^	0.99 ± 0.25 ^§§^
TIMP-1 (sera pg/mL; tissues pg/mg protein)				
Serum	435 ± 175	794 ± 300	2154 ± 477 ****^,††††^	839 ± 463 ^§§§§^
Stomach	592 ± 189	593 ± 121	893 ± 113 **^,††^	631 ± 138 ^§^
Small Intestine	57.4 ± 30.3	54.9 ± 27.1	213.2 ± 62.9 ***^,†††^	124.5 ± 64.7 ^§^
Large Intestine	378 ± 62	470 ± 65	669 ± 50 ****^,†††^	551 ± 74 ^§^

* *p* < 0.05 ** *p* < 0.01 *** *p* < 0.001 **** *p* < 0.001 compared to Control; ^†^
*p* < 0.05 ^††^
*p* < 0.01 ^†††^
*p* < 0.001 ^††††^
*p* < 0.0001 compared to ALA; ^§^
*p* < 0.05 ^§§^
*p* < 0.01 ^§§§^
*p* < 0.001 ^§§§§^
*p* < 0.0001 compared to Mucositis.

**Table 3 antioxidants-11-01930-t003:** Histomorphological scoring of small intestine villus heights (µm).

Groups	Villus Heights
Control	425.56 ± 77.67
ALA	423.64 ± 69.99
Mucositis	351.83 ± 40.22
Mucositis +ALA	399.06 ± 42.69
*p* value	0.183

The *p* value represents the degree of statistical significance of the Kruskal-Wallis test results.

## Data Availability

The data that support the findings of this study are available from the corresponding authors upon reasonable request.

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
