# Peer review of "Protective Effects of Alpha-Lipoic Acid against 5-Fluorouracil-Induced Gastrointestinal Mucositis in Rats"

_antioxidants, 2022, doi:10.3390/antiox11101930_

Round 1
Reviewer 1 Report
In the manuscript by Ceylanli et al., have illustrated the protective role of Alpha-lipoic acid in 5-Fluorouracil-induced gastrointestinal mucositis in rats. Further, the author has given a mechanistic insight, where they have shown that treatment with alpha-lipoic acid increased the SOD and GPx activities in mucositis rats, and thus reduced the damage induced by mucositis both in the stomach and intestine. In general, the manuscript is well written, and all the data are nicely explained. However, I have one suggestion for the manuscript:
Representative images in Figures 3, 4, and 5 are bit small; please along with the X10 images provide an enlarged area from the same image for better interpretation.
Author Response
Reviewer #1 Comments
In the manuscript by Ceylanli et al., have illustrated the protective role of Alpha-lipoic acid in 5-Fluorouracil-induced gastrointestinal mucositis in rats. Further, the author has given a mechanistic insight, where they have shown that treatment with alpha-lipoic acid increased the SOD and GPx activities in mucositis rats, and thus reduced the damage induced by mucositis both in the stomach and intestine. In general, the manuscript is well written, and all the data are nicely explained.
Our Response: We would like to express our sincere thanks for your overall opinion.
However, I have one suggestion for the manuscript: Representative images in Figures 3, 4, and 5 are bit small; please along with the X10 images provide an enlarged area from the same image for better interpretation
Our Response: Based on your suggestion, we have now improved these figures in terms of presentation design, size, and dpi. Also, we have now revised the figure legends. For example, we have replaced the magnification information with a scale bar (bar = 100 µm). Last, we have now enlarged the relevant areas in the figures for better interpretation (bar = 50 µm).
Reviewer 2 Report
The manuscript investigated whether the combination of alpha-lipoic acid (ALA) and 5-fluorouracil (5-FU) therapy had a potential impact on patients with gastroenteritis. They found that ALA had the ability to reverse 5-FU-induced increases in oxidative stress and inflammation-related marker, and to improve the degree of villous damage on gastrointestinal tissue. Unfortunately, I think the author's performance in the figure is not strict enough, because the order of the representative diagram in histopathological staining cannot correspond to the number of the legend, and the author did not display a scale bar on each tissue staining. In addition, the title is recommended to be named a positive sentence that conforms to the results, and also the subtitle of the result should be prioritized in one sentence, not a separate word. Furthermore, if "*" indicates a statistical difference, why is there a P value displayed in Table 3? In view of this, I think this study needs to be partially revised.
Author Response
Reviewer #2 Comments
The manuscript investigated whether the combination of alpha-lipoic acid (ALA) and 5-fluorouracil (5-FU) therapy had a potential impact on patients with gastroenteritis. They found that ALA had the ability to reverse 5-FU-induced increases in oxidative stress and inflammation-related marker, and to improve the degree of villous damage on gastrointestinal tissue.
Our Response: We would like to express our sincere thanks for considering our manuscript.
Unfortunately, I think the author's performance in the figure is not strict enough, because the order of the representative diagram in histopathological staining cannot correspond to the number of the legend, and the author did not display a scale bar on each tissue staining.
Our Response: Based on your suggestion, we have now improved the figures and revised the figure legends. We have also checked Section 3.5 and the associated figure legends and made the required changes.
In addition, the title is recommended to be named a positive sentence that confoms to the results, and also the subtitle of the result should be prioritized in one sentence, not a separate word.
Our Response: Following your suggestion, we have now made some modifications to the title of the manuscript as well as the subtitles in the Results section.
Furthermore, if "*" indicates a statistical difference, why is there a P value displayed in Table 3? In view of this, I think this study needs to be partially revised.
Our Response: We greatly appreciate your attention to this matter. We have now re-evaluated our statistical data. We confirm that the p-value indicated in Table 3 (p = 0.183) is derived from the Kruskal–Wallis test. We believe that the Mann–Whitney U test is no longer necessary. Accordingly, there is no significant differences between the groups. In the light of these results, we have now revised the relevant parts of the main text.
Round 2
Reviewer 2 Report
The revised manuscript can be accepted now.